# Parental migration and psychological well-being of left-behind adolescents in Western Nepal

Madhu Kharel, Shibanuma Akira, Junko Kiriya, Ken Ing Cherng Ong*, Masamine Jimba

Department of Community and Global Health, Graduate School of Medicine, The University of Tokyo, Tokyo, Japan

* kenicong@m.u-tokyo.ac.jp

**Data Availability Statement:** All data files are available in the Figshare Repository at: https://dx.doi.org/10.6084/m9.figshare.12301937.

**Funding:** This study is funded by The University of Tokyo Fund to MJ. The funder had no role in study

## Abstract

### Introduction

International migration is increasing rapidly around the world mostly to obtain a job. International migrant workers usually leave their children back in their country of origin, and among family members, adolescents may experience greater psychological distress from parental separation. However, limited evidence is available on the relationship between parental international migration and psychological well-being of left-behind adolescents. Nepal has a relatively higher and increasing number of international migrants, and this study was conducted to examine the association between parental international migration and the psychological well-being of left-behind adolescents in Nepal.

### Methods

A school-based cross-sectional study was conducted among 626 adolescents in two districts of Western Nepal, where international migration is common. Adolescents were recruited through random sampling. Pre-tested "Strengths and Difficulties Questionnaire" was used to measure their psychological well-being and simple and multiple linear regression were used to examine the association between parental international migration and the psychological well-being of left-behind adolescents.

### Results

Adolescents with none of the parents living abroad were more likely to have higher total difficulties score compared to those with one of the parents living abroad (B: 1.02; 95% CI: 0.18, 1.86; $p = 0.017$). Adolescents with the following factors were more likely to have higher total difficulties score in comparison to their counterparts: adolescents in their late adolescence period, female adolescents, adolescents from ethnicities other than Brahmin and adolescents studying in private schools.

design, data collection and analysis, decision to publish, or preparation of the manuscript.

**Competing interests:** The authors have declared that no competing interests exist.

## Conclusion

In rural districts of Nepal, where international migration is common, adolescents living with the parents were more likely to have poorer psychological well-being compared to those with one of the parents living abroad. Adolescents' adaptation mechanism for the absence of parents for international migration might be explored in the future studies.

## Introduction

International migration is increasing rapidly around the world. In 2019, for example, around 272 million people were estimated to live in a foreign country, accounting for 3.5% of the world's population [1]. Employment is the most common reason for international migration and migrant workers accounted for nearly two-thirds of the world's 258 million international migrants in 2017 [2].

Many migrants go abroad for employment and leave their children in their country of origin under the care of the remaining partner or other family members or relatives. Among them, children and adolescents, who are at the critical stage of growth and development, are anticipated to experience greater psychological distress from parental separation compared to other family members [3]. Precise data on the number of children left-behind is not available, but the number is thought to be in the hundreds of millions [4].

Despite such family problems, international migration is important in Nepal because remittance plays a vital role in the Nepalese economy. It accounted for about 31% of the Gross Domestic Product (GDP) in 2016, and Nepal ranked the second among the top five countries in the world with the highest share of remittance on GDP [5].

Nepal, a low-income country (recently upgraded to a lower-middle-income country [6]) in South Asia, is a labour exporting country. Due to the lack of employment opportunities in the country, people are forced to go abroad for work [7]. According to the National Census of 2011, one-fourth of the households had at least one member living in a foreign country [8]. Labour migration gained popularity among Nepalese after the restoration of democracy in 1990 when Nepal adopted liberal economy and opened up labour migration to countries other than India. Since that time, the number of people going abroad for work has increased significantly. A total of 3,554,683 foreign employment permits were issued between 2008/09-2016/17 for 153 countries by the Department of Foreign Employment [7]. Labour migration will increase in the future as the government is trying to make the labour migration process smooth, safe and transparent. The majority of Nepalese migrant workers leave their children in Nepal when they go abroad. The children are looked after by the remaining family members in Nepal, mostly their grandparents as joint families are still common in Nepal.

As labour migration will not decrease in the coming years in Nepal and other resource limited settings, research has been conducted on the impact of parental migration on the mental health of left-behind children [9–11]. However, a large proportion of these studies were conducted in China and mainly focused on internal migration [4, 12, 13]. Compared to internal migration, international migration of parents could have a much bigger impact on the mental health of children and adolescents left-behind. This is because it can increase geographical distance and limit the frequency of parent-child meeting and communication. Apart from the studies in China, studies have been conducted in some parts of Asia and Africa to see the impact of parental international migration on the mental health of left-behind children and adolescents [9, 10, 14]. However, these studies have produced mixed results, with some

reporting negative effects of parental migration [9, 10, 14] while others reporting no difference in the psychological well-being of adolescents [9, 14]. Moreover, only a few studies have focused on adolescents.

Adolescents are in a crucial period of physical, sexual, and psychological development [3] and they may need more parental support during this period. If the parents are absent during this period, they might face various challenges including psychological distress. However, few studies have focused on them, and more evidence is necessary on how the absence of parents due to international migration could affect the mental health of left-behind adolescents. Therefore, this study examined the association between parental international migration and the psychological well-being of left-behind adolescents in Nepal.

## Methods

### Study design and settings

This study was a school-based, cross-sectional study conducted in two districts of Western Nepal, Baglung and Gulmi. These are basically rural districts. The two districts were selected as the proportion of households with at least one member living in a foreign country in these two districts was among the highest (Gulmi: 54.1% and Baglung: 47.4%) in Nepal [15]. Two municipalities, one from each district, with the high proportion of absent households (Galkot Municipality from Baglung and Musikot Municipality from Gulmi) were selected as the study sites. These two areas were declared municipalities by the government in March 2017, after merging several Village Development Committees.

### Participants

The participants in this study were adolescents studying in grades from 8 to 10, and between the age of 11 and 17 years with both parents alive. Adolescents were excluded when they had severe mental health illnesses, or they were absent from school on the day of data collection, their parents were migrating internally, or divorced. Sample size was calculated using OpenEpi software based on the findings from a similar study in Angola in Africa (a lower-middle income country as Nepal) as studies from Nepal or other Asian countries were not found which had relevant information on adolescents' psychological well-being to calculate the sample size for the study. In Angola, the adolescents living with both parents had mean psychological well-being score of 13.0 and standard deviation (SD) of 5.6. These figures for the adolescents with both parents away were 15.1 and 5.9, respectively [14]. The design effect was calculated to be 2.56 assuming the average class size of 40 and the intra-class correlation coefficient of 0.04 [16]. Power was set at 80% and the significance level was set at 5%. Considering a non-response rate of 10%, the minimum required sample size was 685.

### Sampling procedure

Four schools (two public and two private) per municipality were selected randomly from the list of public secondary schools and private secondary schools in each municipality. In total, four public schools and four private schools were selected for the survey. Adolescents in Grades 8, 9 and 10 filled the questionnaire. Grades are divided into two or more sections (e.g. 9A, 9B and so on for Grade 9) when the school have either large number of students or different teaching languages (Nepali or English) or different majors (such as Accounting, Agriculture, Computer, Education, Optional Maths) in a grade. Two randomly selected sections were included from each grade of a school when the particular grade in that school had more than two sections. In this study, 3 schools had 3 sections in Grades 9 and 10.

## Measures

In this study, the exposure variable was parental migration status and the outcome variable was psychological well-being of adolescents. Information on adolescents' perceived relationship with their primary caretaker and other socio-demographic variables were also collected.

**Parental migration status.** The main exposure variable in this study was parental migration status. Adolescents were asked if their father and/ or mother was living abroad. The responses were categorised as none of the parents living abroad, only the father living abroad, only the mother living abroad and both parents living abroad.

**Relationship with primary caretaker.** Adolescents provided response on their perceived relationship with their primary caretaker. They had four options to choose from: "Very good", "Satisfactory", "Poor" and "Very poor".

**Psychological well-being of adolescents.** Psychological well-being of adolescents was the outcome variable, and it was measured using the Strengths and Difficulties Questionnaire (SDQ). It is widely used around the world to assess mental well-being of children and is translated into more than 80 languages, including Nepali [17, 18]. It is a 25 item scale consisting of 5 sub-scales: emotional symptoms, conduct problems, hyperactivity/ inattention, peer relationships problem and prosocial behaviour [19]. The five sub-scales contain five questions each. The first 4 sub-scales yield total difficulties score which ranges from 0 to 40. Use of broader scale score is preferred over the sub-scale scores in community samples with low-risk of mental health difficulties [20]. Therefore, the total difficulties score was used to measure the psychological well-being of adolescents in this study as done by similar studies in the past [14, 21]. Higher total difficulties score indicates poorer psychological well-being.

**Sociodemographic characteristics.** Sociodemographic variables included age, gender, ethnicity, religion, parents' educational status, economic status (wealth quintile) and school type. Age was categorised into "Early adolescence" (10–14 years) and "Late adolescence" (15–19) based on the definition given by the World Health Organization (WHO) [3].

## Data collection

Self-administered questionnaire was used to collect information on exposure, outcome and other variables. The students filled out the questionnaire in their classrooms, in the presence of the researcher, who provided instructions on how to fill the questionnaire in the beginning and was available throughout the session to clarify any confusion or questions. It took around 40–45 minutes to complete the questionnaire. Data collection was facilitated through coordination with respective municipalities, their education sections and selected schools. The questionnaire was pre-tested among adolescents in similar settings. The instructions for filling the questionnaire were made clearer based on the results of the pre-test. Data were collected in August and September 2019.

## Data analysis

One-way Analysis of Variance (ANOVA) was used to compare the mean total difficulties score between adolescent groups in different categories of each variable. Simple and multiple linear regression analyses were performed to examine the association between independent variables and psychological well-being of adolescents. Variance Inflation Factor (VIF) was calculated to check multicollinearity between independent variables. All analyses were performed using Stata version 15.1 (College Station, TX: StataCorp LLC).

### Ethics

This study was approved by the Research Ethics Committee of The University of Tokyo (Serial Number: 2019067NI) in Japan as well as the Nepal Health Research Council (Reg. no. 615/2019) in Nepal. Participation in this study was voluntary. Written informed assent was collected from the adolescents and written informed consent was obtained from their guardians before data collection. The questionnaire did not include any personal identifiable information.

## Results

A total of 763 adolescents completed the survey. All the children present on the days of the survey completed the questionnaire. However, for analysis, those with divorced, deceased and internal migrant parents were excluded. Therefore, final analysis was conducted among 626 eligible adolescents.

### General characteristics of adolescents

Table 1 shows the demographic characteristics of the adolescents (n = 626). Their mean age was 14.3 years [Standard Deviation (SD) 1.2] and more than half of them were in their early adolescence (10–14 years). The proportion of female adolescents (52.1%) was slightly higher than that of male adolescents (47.9%). Majority of the adolescents followed Hindu religion. Chhetri was the largest ethnic group accounting for nearly one-third of the adolescents. More than two-thirds of the adolescents attended public schools. Majority of the adolescents said they had a very good relationship with their primary caretakers. Very few respondents said they had poor (n = 1) and very poor (n = 2) relationship with their primary caretaker. Therefore, the three categories "Satisfactory", "Poor" and "Very Poor" were collapsed into a new category—"Not very good".

### Parental migration status

More than half of the adolescents were living with both parents. One-fourth of the adolescents had one of their parents living abroad and around 1 in 10 adolescents had both of their parents living abroad. Among those with one of the parents living abroad, very few adolescents (n = 4) had their mother living abroad (not shown in the table). So, the two categories—"only the father living abroad" and "only the mother living abroad" were collapsed to form a new category—"one of the parents living abroad".

### Total difficulties score

The tool used to collect information on the psychological well-being of adolescents had acceptable reliability (Cronbach's alpha = 0.69). Table 2 shows comparison of the mean total difficulties score between adolescent groups in different categories in each independent variable. Adolescents with both parents living abroad had higher total difficulties score compared to those with one of the parents living abroad (pairwise comparisons using Bonferroni correction not shown in the table). The following groups had higher total difficulties score compared to their counterparts: older adolescents, female adolescents, adolescents from ethnicities other than Brahmin and those studying in private schools. No significant difference was observed in the mean total difficulties score for religion, parents' education, wealth quintile and children's relationship with their primary caretaker.

**Table 1. General characteristics of adolescents (n = 626).**

| Variables | | n | %* |
|---|---|---:|---:|
| **Age (years) Mean (SD)** | | | 14.3 (1.2) |
| **Age group** | | | |
| | Early adolescence (10–14 years) | 354 | 56.6 |
| | Late adolescence (15–19 years) | 272 | 43.5 |
| **Gender** | | | |
| | Male | 300 | 47.9 |
| | Female | 326 | 52.1 |
| **Type of school** | | | |
| | Public | 457 | 73.0 |
| | Private | 169 | 27.0 |
| **Religion** | | | |
| | Hindu | 607 | 97.0 |
| | Buddhist | 11 | 1.8 |
| | Christian | 6 | 1.0 |
| | Muslim | 1 | 0.2 |
| | Other | 1 | 0.2 |
| **Ethnicity** | | | |
| | Brahmin | 106 | 16.9 |
| | Chhetri | 234 | 37.4 |
| | Janajati | 173 | 27.6 |
| | Dalit | 112 | 17.9 |
| | Muslim | 1 | 0.2 |
| **Father's education** | | | |
| | Did not complete primary level | 113 | 18.1 |
| | Completed primary or secondary level | 424 | 67.7 |
| | Completed higher than secondary level | 89 | 14.2 |
| **Mother's education** | | | |
| | Did not complete primary level | 178 | 28.4 |
| | Completed primary or secondary level | 385 | 61.5 |
| | Completed higher than secondary level | 63 | 10.1 |
| **Parental migration status** | | | |
| | None of the parents living abroad | 383 | 61.2 |
| | One of the parents living abroad | 166 | 26.5 |
| | Both parents living abroad | 77 | 12.3 |
| **Relationship with primary caretaker** | | | |
| | Very good | 584 | 93.3 |
| | Satisfactory | 39 | 6.2 |
| | Poor | 1 | 0.2 |
| | Very poor | 2 | 0.3 |

SD, standard deviation.

* Total might not add up to 100 due to rounding off.

## Association between parent's migration status and psychological well-being of left-behind adolescents

Table 3 shows the results of simple linear regression analysis for the association of each independent variable with psychological well-being of adolescents, without controlling for other

**Table 2. Total difficulties score by general characteristics (n = 626).**

| Variables | n | Mean | SD | p-value |
|---|---|---|---|---|
| **All adolescents** | 626 | 9.6 | 4.7 | |
| **Age group** | | | | **0.001** |
| Early adolescence | 354 | 9.1 | 4.6 | |
| Late adolescence | 272 | 10.3 | 4.7 | |
| **Gender** | | | | **0.046** |
| Male | 300 | 9.3 | 4.4 | |
| Female | 326 | 10.0 | 4.9 | |
| **Ethnicity** | | | | **0.002** |
| Brahmin | 106 | 8.3 | 4.3 | |
| Chhetri and others | 520 | 9.9 | 4.7 | |
| **Religion** | | | | 0.992 |
| Hindu | 607 | 9.6 | 4.7 | |
| Buddhist and others | 19 | 9.6 | 4.9 | |
| **Type of school** | | | | **<0.001** |
| Public | 457 | 9.2 | 4.3 | |
| Private | 169 | 10.9 | 5.4 | |
| **Father's education** | | | | 0.387 |
| Lower than primary level | 113 | 9.8 | 3.9 | |
| Completed primary or secondary level | 424 | 9.7 | 4.8 | |
| Completed higher than secondary level | 89 | 9.0 | 5.2 | |
| **Mother's education** | | | | 0.638 |
| Lower than primary level | 178 | 9.7 | 4.3 | |
| Completed primary or secondary level | 385 | 9.7 | 4.8 | |
| Completed higher than secondary level | 63 | 9.1 | 5.3 | |
| **Wealth quintile** | | | | 0.576 |
| Quintile 1 | 134 | 9.3 | 4.0 | |
| Quintile 2 | 115 | 9.3 | 4.5 | |
| Quintile 3 | 126 | 9.8 | 5.0 | |
| Quintile 4 | 133 | 10.1 | 4.9 | |
| Quintile 5 | 118 | 9.8 | 5.0 | |
| **Parental migration status** | | | | **0.025** |
| None of the parents living abroad | 383 | 9.7 | 4.7 | |
| One of the parents living abroad | 166 | 9.0 | 4.3 | |
| Both parents living abroad | 77 | 10.8 | 5.3 | |
| **Relationship with primary caretaker** | | | | 0.375 |
| Very good | 584 | 9.6 | 4.6 | |
| Not very good | 39 | 10.3 | 5.4 | |

Results of one-way analysis of variance (ANOVA).

independent variables. In simple linear regression, the following factors were associated with the psychological well-being of adolescents: parental migration status, age, gender, ethnicity and school type. Adolescents with both parents living abroad were more likely to have higher total difficulties score compared to those with one of the parents living abroad (B: 1.74; 95% CI: 0.48, 3.00; p = 0.007).

Table 4 presents the results of multiple linear regression analysis for the association of each independent variable with psychological well-being of adolescents, while controlling for other

**Table 3. Factors associated with the psychological well-being of adolescents (simple linear regression analysis) (n = 626).**

| Variables | B | 95% CI | p-value |
|---|---|---|---|
| **Age group** (reference: early adolescence) | | | |
| Late adolescence | 1.21 | 0.47, 1.95 | **0.001** |
| **Gender** (reference: male) | | | |
| Female | 0.75 | 0.01, 1.48 | **0.046** |
| **Ethnicity** (reference: Brahmin) | | | |
| Chhetri and others | 1.56 | 0.58, 2.53 | **0.002** |
| **Religion** (reference: Hindu) | | | |
| Buddhist and others | -0.01 | -2.16, 2.14 | 0.992 |
| **Type of school** (reference: public) | | | |
| Private | 1.79 | 0.97, 2.60 | **<0.001** |
| **Father's education** (reference: lower than primary level) | | | |
| Completed primary or secondary level | -0.74 | -1.05, 0.90 | 0.881 |
| Completed higher than secondary level | -0.79 | -2.10, 0.51 | 0.232 |
| **Mother's education** (reference: lower than primary level) | | | |
| Completed primary or secondary level | -0.00 | -0.84, 0.83 | 0.998 |
| Completed higher than secondary level | -0.59 | -1.94, 0.76 | 0.390 |
| **Wealth quintile** (reference: quintile 1) | | | |
| Quintile 2 | 0.27 | -1.14, 1.20 | 0.964 |
| Quintile 3 | 0.50 | -0.64, 1.64 | 0.390 |
| Quintile 4 | 0.83 | -0.30, 1.20 | 0.149 |
| Quintile 5 | 0.49 | -0.68, 1.65 | 0.413 |
| **Parental migration status** (reference: one of the parents living abroad) | | | |
| None of the parents living abroad | 0.68 | -0.17, 1.53 | 0.117 |
| Both parents living abroad | 1.74 | 0.48, 3.00 | **0.007** |
| **Relationship with the primary caretaker** (reference: very good) | | | |
| Not very good | 0.66 | -0.81, 2.13 | 0.375 |

independent variables. The same variables as in simple linear regression analysis were associated with the psychological well-being of adolescents in multiple linear regression analysis. Adolescents living with both parents were more likely to have higher total difficulties score compared to those with one of the parents living abroad (B: 1.02; 95% CI: 0.18, 1.86; p = 0.017). Adolescents with both parents living abroad were also more likely to have higher total difficulties score than their peers with one of the parents living abroad (B: 1.28; 95% CI: -0.04, 2.60; p = 0.057). However, this association was not significant at 5% level of significance. Adolescents in their late adolescence period (B: 1.05; 95% CI: 0.32, 1.78; p = 0.005), female adolescents (B: 0.80; 95% CI: 0.05, 1.55; p = 0.037), adolescents from ethnicities other than Brahmin (B: 1.33; 95% CI: 0.30, 2.36; p = 0.011) and adolescents studying in private schools (B: 1.77; 95% CI: 0.73, 2.81; p = 0.001) were more likely to have higher total difficulties score in comparison to their counterparts. Multi-collinearity was not observed in the model as the VIF for all the variables was below 2.35.

## Discussion

This study has the following major findings. Adolescents living with both parents were more likely to have higher total difficulties score and thus their psychological well-being was poorer compared to those with one of the parents living abroad. Older adolescents, female

**Table 4. Factors associated with the psychological well-being of adolescents (multiple linear regression analysis) (n = 626).**

| Variables | B | 95% CI | p-value |
|---|---|---|---|
| **Age group** (reference: early adolescence) | | | |
| Late adolescence | 1.05 | 0.32, 1.78 | **0.005** |
| **Gender** (reference: male) | | | |
| Female | 0.80 | 0.05, 1.55 | **0.037** |
| **Ethnicity** (reference: Brahmin) | | | |
| Chhetri and others | 1.33 | 0.30, 2.36 | **0.011** |
| **Religion** (reference: Hindu) | | | |
| Buddhist and others | -0.33 | -2.35, 1.69 | 0.747 |
| **Type of school** (reference: public) | | | |
| Private | 1.77 | 0.73, 2.81 | **0.001** |
| **Father's education** (reference: lower than primary level) | | | |
| Completed primary or secondary level | -0.23 | -1.24, 0.78 | 0.655 |
| Completed higher than secondary level | -1.00 | -2.56, 0.55 | 0.204 |
| **Mother's education** (reference: lower than primary level) | | | |
| Completed primary or secondary level | -0.09 | -1.01, 0.84 | 0.849 |
| Completed higher than secondary level | 0.16 | -1.58, 1.91 | 0.856 |
| **Wealth quintile** (reference: quintile 1) | | | |
| Quintile 2 | 0.21 | -0.88, 1.31 | 0.701 |
| Quintile 3 | 0.32 | -0.80, 1.45 | 0.571 |
| Quintile 4 | 0.43 | -0.83, 1.69 | 0.504 |
| Quintile 5 | -0.03 | -1.40, 1.34 | 0.966 |
| **Parental migration status** (reference: one of the parents living abroad) | | | |
| None of the parents living abroad | 1.02 | 0.18, 1.86 | **0.017** |
| Both parents living abroad | 1.28 | -0.04, 2.60 | 0.057 |
| **Relationship with the primary caretaker** (reference: very good) | | | |
| Not very good | 0.55 | -1.04, 2.14 | 0.494 |

adolescents, adolescents from ethnicities other than Brahmin, and adolescents studying in private school were more likely to have poorer mental well-being.

This study found that adolescents living with both parents were more likely to have poorer psychological well-being compared to those with one of the parents living abroad. Parental migration is believed to have negative impact on the mental health of left-behind children and adolescents [4]. However, past studies also have shown that parental migration may not always be associated with poorer mental health among left-behind children and adolescents, and circumstances of parental migration and living arrangements for the left-behind children and adolescents might make a difference. For example, in Africa, no difference was observed for mental well-being between adolescents living with both parents and those with father or both parents living abroad in Ghana and Nigeria; however, adolescents with one or both parents living abroad had poorer psychological well-being compared to those living with both parents in Angola [14]. In our study, almost all of the adolescents with one of the parents living abroad had their father away and mother was there to take care of them. Therefore, they might have a sense of financial security from their father's earning as well as a sense of emotional support from their mother's presence at home. The majority of the migrant parents of the target population were living in Japan, one of the high-income countries and it might be a matter of pride for their children [22]. However, wealth quintile was not associated with psychological well-being of adolescents in this study. Joint families are quite common in Nepal, especially in rural

areas, and the children are usually looked after by their grandparents. As the migrant parents mostly leave their children under the care of their grandparents, the left-behind children and adolescents might not have experienced much difference in the love and care they received before and after their parents migrate.

Generally, the school-going adolescents in Nepal support their families with household work though they are not engaged in direct income-generating activities. Agriculture is the primary occupation of the majority of the families in the study area. Families without foreign income may have to work more in the field to sustain their livelihoods. Accordingly, adolescents in such families might have to work extra hours to support their families. This extra burden on adolescents can affect their psychological well-being.

As anticipated, adolescents with both parents living abroad tended to show higher total mean difficulties score compared to those with none or one of the parents living abroad. However, this difference was not significant at 5% level of significance. Since the number of adolescents with both parents living abroad was low, this study might have failed to detect this difference.

In this study, adolescents in their late adolescence were more likely to have poorer mental well-being compared with those in their early adolescence. This concurs with previous studies in Nepal and Spain, too [23, 24]. This might be because the older adolescents are more likely to report lower life satisfaction, lower perceived family support, lower quality of communication with parents and increased school pressure [24, 25]. Being female was negatively associated with mental well-being in this study. This finding is consistent with results from previous studies in Nepal and other countries [23, 26–28]. In a multi-country study from Europe and North America, female adolescents had lower levels of satisfaction with life than their male counterparts, and they were more likely to report multiple health complaints [25]. Adolescents from ethnicities other than Brahmin were more likely to have poorer psychological well-being compared to Brahmin adolescents in this study. The caste-based hierarchy which is still existing in Nepalese society might have played a role in this ethnic difference in mental well-being [29, 30]. Studying in private school was associated with lower mental well-being among adolescents. School is the second home of children and the school environment plays a vital role in their development and psychological well-being. Private schools are generally believed to provide better quality education compared to public schools in Nepal and parents prefer to send their children to private schools [31]. However, private schools may not always be conducive to the emotional well-being of their students as observed in a study in the US [32]. This might be because of stricter school rules and higher academic pressure in private schools compared to public schools [33].

The finding from this study should be interpreted in consideration with the following limitations. First, this study used only the self-report version of SDQ although it would have been ideal to get information from different sources such as parents and teachers. However, it has been established from the past studies that the self-report version of SDQ could be used for the group of adolescents included in this study [34]. Second, this study used a self-administered questionnaire and some missing responses were observed despite clear instructions while filling the questionnaire. Those respondents had to be excluded from the analysis. Third, this study could not exclude adolescents with divorced parents, internal migrant parents and deceased parents during the survey due to ethical reasons. These could have caused selection bias. Fourth, the characteristics of primary caretakers, other than their relationship with the adolescents, were not captured in this study. Next, sub-group analysis could not be performed for adolescents' gender and age group as the study will not have enough power to detect the difference due to small sample size. Another limitation of this study is that the number of adolescents with both parents living abroad was relatively small compared to those living with

both parents. Finally, due to the cross-sectional design of this study, causality cannot be established.

Despite these limitations, this study is one of the few studies examining the association between parental international migration and the psychological well-being of left-behind adolescents. This study also included information on adolescents' perceived relationship with their primary caretakers. The findings from this study might be useful in understanding the association between parental international migration and the psychological well-being of left-behind adolescents in settings where parental international migration is common.

This study highlights the need to prioritize adolescents with none or both parents living abroad, older adolescents, female adolescents, adolescents from ethnicities other than Brahmin and those attending private schools while enacting policies and interventions to promote the psychological well-being of adolescents.

## Conclusion

In rural districts of Nepal, where international migration is common, adolescents living with the parents were more likely to have a poorer psychological well-being compared to those with one of the parents living abroad. Future studies could explore adolescents' adaptation mechanism for the absence of parents due to international migration.

## Supporting information

**S1 File. STROBE checklist.**
(DOC)

**S2 File. General characteristics by parental migration status.**
(DOCX)

**S3 File. Coding of study variables.**
(DOCX)

## Acknowledgments

We would like to express our heartfelt thanks to all adolescents for taking part in the study and to their guardians for allowing them to participate. Authorities in the two municipalities and the participating schools also deserve deep appreciation for their cooperation and support during data collection. We would also like to acknowledge and thank Pushkar Raj Silwal and Vishnu Prasad Sapkota for their inputs during proposal development and data analysis.

## Author Contributions

**Conceptualization:** Madhu Kharel, Ken Ing Cherng Ong, Masamine Jimba.

**Data curation:** Madhu Kharel, Shibanuma Akira.

**Formal analysis:** Madhu Kharel, Shibanuma Akira, Ken Ing Cherng Ong, Masamine Jimba.

**Funding acquisition:** Masamine Jimba.

**Investigation:** Madhu Kharel, Shibanuma Akira, Junko Kiriya, Ken Ing Cherng Ong, Masamine Jimba.

**Methodology:** Madhu Kharel, Shibanuma Akira, Junko Kiriya, Ken Ing Cherng Ong, Masamine Jimba.

**Project administration:** Madhu Kharel.

**Resources:** Madhu Kharel, Junko Kiriya, Masamine Jimba.

**Software:** Madhu Kharel, Shibanuma Akira.

**Supervision:** Shibanuma Akira, Junko Kiriya, Ken Ing Cherng Ong, Masamine Jimba.

**Validation:** Madhu Kharel, Shibanuma Akira, Junko Kiriya, Ken Ing Cherng Ong, Masamine Jimba.

**Writing – original draft:** Madhu Kharel.

**Writing – review & editing:** Shibanuma Akira, Junko Kiriya, Ken Ing Cherng Ong, Masamine Jimba.

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
