## [Decision Letter · Decision Letter 0]

9 Sep 2020

PONE-D-20-14440

Parental migration and psychological well-being of left-behind adolescents in Western Nepal

PLOS ONE

Dear Dr. Ong,

Thank you for submitting your manuscript to PLOS ONE. After careful consideration, we feel that it has merit but does not fully meet PLOS ONE’s publication criteria as it currently stands. Therefore, we invite you to submit a revised version of the manuscript that addresses the points raised during the review process.

The manuscript has been evaluated by two reviewers, and their comments are available below. You will see the reviewers have commented on the potential impact of your work and interest it will attract once published. However, they have also raised a number of concerns that should be addressed before the manuscript can be accepted.

Please note that as per our publication criteria, PLOS ONE requires that all experiments, statistics and other analyses are performed to a high technical standard, described in sufficient detail and adhere to appropriate reporting guidelines and community standards. Conclusions must be presented in an appropriate fashion and be supported by the data (Please see http://journals.plos.org/plosone/s/criteria-for-publication).

The reviewers have requested further details regarding the statistical analysis and requested the further variables be included in your analysis. You should consider the requests made by the reviewers to ensure that the data presented in the manuscript support the conclusions drawn.

We look forward to receiving your revised manuscript.

Kind regards,

George Vousden

Senior Editor

PLOS ONE

Journal Requirements:

Reviewers' comments:

Reviewer's Responses to Questions

**Comments to the Author**

1. Is the manuscript technically sound, and do the data support the conclusions?

Reviewer #1: Yes

Reviewer #2: Partly

2. Has the statistical analysis been performed appropriately and rigorously? 

Reviewer #1: Yes

Reviewer #2: No

3. Have the authors made all data underlying the findings in their manuscript fully available?

Reviewer #1: No

Reviewer #2: Yes

4. Is the manuscript presented in an intelligible fashion and written in standard English?

Reviewer #1: Yes

Reviewer #2: Yes

5. Review Comments to the Author

Reviewer #1: Many thanks for the opportunity to review this paper on the mental well-being of left-behind adolescents in Nepal. I apologise for the delay in submitting my review.

The research has been carried out to a high standard. The paper covers an important topic which has not received much attention to date in Nepal. The methods are rigorous and the authors have described them in detail. The conclusions are sound and of interest to a wide, global health audience.

I have some minor comments below for the authors' consideration.

Results, 1st paragraph: It may be clearer if you state that all children present on the days of the surveys completed the questionnaire. However, for analysis, those with divorced/deceased parents were excluded. At the moment it is slightly unclear from the way you have described it.

Results, Table 1. I would find it helpful to see these demographic characteristics according to parental migrant status. Would it be possible to add to this table (or in an additional table) the distribution of these characteristics by (a) whole sample; (b) one-parent migrant; (c) both parent migrants; (d) non-migrant?

Results, Table 3. It is interesting that you chose 1-migrant-parent as the reference group. PLease can you explain rationale for this? I would imagine it would make more sense to use non-migrant parent group as the reference, because it is the largest group and also it is most relevant to the overall research question - i.e. do children with migrnat parents have worse outcomes than those with no migrant parents. It is slightly more difficult to interpret when the reference category is 1-parent, and you are comparing those to either no-migrant or 2-parents.

Results, Table 3. I also cannot understand why in the simple linear regression, the order of scores changes. In Table 2, 1-parent group has the lowest SDQ scores, followed by non-migrant, and finally 2-parents. In Table 3, this has changed to lowest risk in non-migrant, followed by 1-parent, followed by 2-parents. What has changed, given that in the simple regression model you are (presumably) not controlling for any other factors yet?

Reviewer #2: Dear Authors,

I read your work with great interest and I commend you on focusing on left behind children on Nepal. Your study has a lot of potential to provide more insights into the existing literature on the psychological well-being left behind children. Below I provide you with comments to improve your work.

1. On page 4, line7, your paper argues that most studies focus on the left behind children in China in an internal migration context. However, there are now more studies on the psychological well-being of left behind children in Africa, Latin America and other parts of Asia (the Philippines, Indonesia). It is important that you acknowledge and get insights from these quantitative studies, which will help you in improving the choice of variables for your analysis and improving the key conclusions you can draw from this analysis.

2. In the Methods section,

a. the use of psychological data on the African adolescents is neither properly explained nor very well justified. It is not particularly clear for what purpose you use this data and what insight it can bring to the study, as well on what basis Nepalese and African Adolescents can be compared. Note: it is very important that you mention the exact country in Africa that this data represents as Africa is a continent covering many countries. If you are comparing continental average data, it is important to explicit about that.

b. The methods section should be explicit about what kind of information were collected in the survey.

c. It is important to add a supplementary table on the coding of the socioeconomic and demographic variables.

d. Because you are focusing on the left behind children, it is crucial that you account for the characteristics of the caregiver: is the caregiver employed, other than relationship with the caregiver. It is also important to control for the economic activities of these children (see #3 comment for more explanation). Please also consider variables such as household size if it is available.

e. In the analyses strategy, it is important to conduct further analyses to check the robustness of your results. Contrary to findings for Ghanaian and Angolan left behind children, your study finds that adolescents living with both their parents have lower psychological well-being. I suggest you conduct separate analyses for girls and boys to see whether this result persists. You can also conduct separate analyses for early adolescent and late adolescent groups as the effect of parental migration could differ between these groups.

3. To explain the results you find, it might be wise to look into childhood and youth studies, which study these groups in non-migratory contexts. It is also worthwhile to include some background information about Nepal’s socioeconomic and adolescent labor conditions to understand the emigration context better. That could for instance help to nuance why adolescents living with both parents could be in a worse situation. It could be that they may themselves need to work to support their families’ livelihood.

6. PLOS authors have the option to publish the peer review history of their article (what does this mean?). If published, this will include your full peer review and any attached files.

Reviewer #1: No

Reviewer #2: No

---

## [Author Response · Author response to Decision Letter 0]

21 Oct 2020

We have attached the response as a file named "Response to Reviewers".

---

## [Decision Letter · Decision Letter 1]

25 Nov 2020

PONE-D-20-14440R1

Parental migration and psychological well-being of left-behind adolescents in Western Nepal

PLOS ONE

Dear Dr. Ong,

Thank you for submitting your manuscript to PLOS ONE. After careful consideration, we feel that it has merit but does not fully meet PLOS ONE’s publication criteria as it currently stands. Therefore, we invite you to submit a revised version of the manuscript that addresses the points raised during the review process.

We look forward to receiving your revised manuscript.

Kind regards,

Gracia Fellmeth

Academic Editor

PLOS ONE

Additional Editor Comments (if provided):

Dear Associate Professor Ong,

Many thanks for submitting your revised manuscript. I have taken on the role of editor for your manuscript, having previously reviewed your original submission (my comments were those of "Reviewer 1" in the original submission).

I have now had a chance to read your revised version and feel it is much improved. Thank you for addressing the points raised by myself and the second reviewer. Both the second reviewer and I feel there are still some issues remaining which need to be addressed before this is suitable for publication. Please see these listed below.

We look forward to receiving another revised version.

Sincerely,

Dr Gracia Fellmeth

(Guest Editor)

Reviewers' comments:

Reviewer's Responses to Questions

**Comments to the Author**

1. If the authors have adequately addressed your comments raised in a previous round of review and you feel that this manuscript is now acceptable for publication, you may indicate that here to bypass the “Comments to the Author” section, enter your conflict of interest statement in the “Confidential to Editor” section, and submit your "Accept" recommendation.

Reviewer #2: (No Response)

2. Is the manuscript technically sound, and do the data support the conclusions?

Reviewer #2: Partly

3. Has the statistical analysis been performed appropriately and rigorously? 

Reviewer #2: (No Response)

4. Have the authors made all data underlying the findings in their manuscript fully available?

Reviewer #2: Yes

5. Is the manuscript presented in an intelligible fashion and written in standard English?

Reviewer #2: Yes

6. Review Comments to the Author

Reviewer #2: Dear Authors,

In general the motivation for such kind of research is nicely presented. The text is written in a clear and precise way.

Having said this, there are four major issues that need your attention.

1. It is not clear why you call Table 3 a simple linear regression analyses and Table 4 a multiple linear regression analyses. A simple linear regression is when you only have one variable in your analyses. By this logic, Table 3 is rather a multiple linear regression table. It is also not clear why and how Table 3 and Table 4 differ. I can see that the regression results on the parental migration status variable are not the same in the two tables even though the control variables used are the same. Table 4 shows that children living with both parents in Nepal have a higher SDQ compared to those with one of their parents abroad whereas Table 3 shows that an increased SDQ for children with both parents abroad. What is your explanation for these two opposite findings?

2. It might be worth discussing whether adolescents girls' and boys' SDQs respond differently due to their parent(s)' migration. To do that my suggestion is to run separate regressions for the girls' and boys' sub-samples. Alternatively, you may choose to see the interaction effect of gender and the parental migration status variable. This could give your analyses more nuance.

3. Your conclusion focuses a lot on general discussions on other control variables. It is best if your conclusion dedicates more time contemplating on the result on the parental migration status variable by using some contextual information to explain the result.

4. Suggestion: if you make the conclusion section succinct, then you can have more space to include some background information about parental migration in Nepal either in the introduction section or as a separate section and come back to this in the conclusion section.

7. PLOS authors have the option to publish the peer review history of their article (what does this mean?). If published, this will include your full peer review and any attached files.

Reviewer #2: No

---

## [Author Response · Author response to Decision Letter 1]

21 Dec 2020

Response to Reviewer #2: 

1. It is not clear why you call Table 3 a simple linear regression analyses and Table 4 a multiple linear regression analyses. A simple linear regression is when you only have one variable in your analyses. By this logic, Table 3 is rather a multiple linear regression table. It is also not clear why and how Table 3 and Table 4 differ. I can see that the regression results on the parental migration status variable are not the same in the two tables even though the control variables used are the same. Table 4 shows that children living with both parents in Nepal have a higher SDQ compared to those with one of their parents abroad whereas Table 3 shows that an increased SDQ for children with both parents abroad. What is your explanation for these two opposite findings?

We call Table 3 a simple linear regression analyses because in Table 3, we present the results of bivariate analyses between each independent variable and the dependent variable (i.e. psychological well-being), without controlling for other independent variables. We call table 4 a multiple linear regression analyses because we have controlled for other independent variables while looking at the association between each independent variable and the dependent variable. [UCLA, 2020] 

Table 4 is different from Table 3 in that the Table 3 presents the results of a crude association between each independent variable and the dependent variable while Table 4 shows the results after controlling for potential confounders.

Results from the simple linear regression analyses (Table 3) show that adolescents living with both parents tend to have a higher SDQ compared to those with one of their parents living abroad. However, when we controlled for potential confounders (Table 4), adolescents living with both parents tend to have a higher SDQ compared to those with one of their parents abroad. Therefore, the different results are due to confounding. 

We have removed Table 3 and presented only Table 4 (as Table 3 in the revised manuscript).

Reference:

UCLA Institute for Digital Research and Education Statistical Consulting. Regression with STATA Chapter 1 - Simple and multiple regression [Internet]. California: UCLA Institute for Digital Research and Education; [cited: 2020 Dec 19]. Available from: https://stats.idre.ucla.edu/stata/webbooks/reg/chapter1/regressionwith-statachapter-1-simple-and-multiple-regression/

2. It might be worth discussing whether adolescents girls' and boys' SDQs respond differently due to their parent(s)' migration. To do that my suggestion is to run separate regressions for the girls' and boys' sub-samples. Alternatively, you may choose to see the interaction effect of gender and the parental migration status variable. This could give your analyses more nuance.

Thank you for your suggestion. We ran multiple regression analysis including the interaction term for gender and parental migration status. However, the interaction term was not statistically significant. Therefore, we did not include it in the final model.

3. Your conclusion focuses a lot on general discussions on other control variables. It is best if your conclusion dedicates more time contemplating on the result on the parental migration status variable by using some contextual information to explain the result.

Thank you for your suggestion. In the revised manuscript, we have discussed more about the parental migration status variable, and shortened the discussion on other control variables.

Please refer to page: 16-18, line: 238-288.

4. Suggestion: if you make the conclusion section succinct, then you can have more space to include some background information about parental migration in Nepal either in the introduction section or as a separate section and come back to this in the conclusion section.

Thank you for your suggestion. We have separated the conclusion section to make it clear. We have also added background information about parental migration in Nepal in the introduction section.

Conclusion Section:

(Page 19, Line: 314-318)

Conclusion

In rural districts of Nepal, where international migration is common, adolescents living with the parents were more likely to have a poorer psychological well-being compared to those with one of the parents living abroad. Future studies could explore adolescents' adaptation mechanism for the absence of parents due to international migration. 

Introduction Section:

(Page: 4, Line: 56-58)

"Nepal, a low-income country (recently upgraded to a lower-middle-income country [6]) in South Asia, is a labour exporting country. Due to the lack of employment opportunities in the country, people are forced to go abroad for work [7]."

(Page: 4, Line: 66-68)

The majority of Nepalese migrant workers leave their children in Nepal when they go abroad. The children are looked after by the remaining family members in Nepal, mostly their grandparents as joint families are still common in Nepal.

(Page: 5, Line: 91) 

This study was a school-based, cross-sectional study conducted in two districts of Western Nepal, Baglung and Gulmi. These are basically rural districts.

(Page: 6; Line: 96-97)

Two municipalities, one from each district, with the high proportion of absent households (Galkot Municipality from Baglung and Musikot Municipality from Gulmi) were selected as the study sites. These two areas were declared municipalities by the government in March 2017, after merging several Village Development Committees.

---

## [Editor Report · Decision Letter 2]

5 Jan 2021

PONE-D-20-14440R2

Parental migration and psychological well-being of left-behind adolescents in Western Nepal

PLOS ONE

Dear Dr. Ong,

Thank you for submitting your manuscript to PLOS ONE. After careful consideration, we feel that it has merit but does not fully meet PLOS ONE’s publication criteria as it currently stands. Therefore, we invite you to submit a revised version of the manuscript that addresses the points raised during the review process.

We look forward to receiving your revised manuscript.

Kind regards,

Gracia Fellmeth

Academic Editor

PLOS ONE

Additional Editor Comments (if provided):

Dear Authors,

Many thanks for submitting your revised manuscript. I am happy that you have addressed all of the reviewers' comments. However, I ask if you can make one final change: in response to one of the reviewer comments, you removed what was previously Table 3 (simple linear regression). In the previous version, it was not clear that Table 3 was indeed a simple regression as it appeared that other factors were being controlled for. However, given your response and clarification, I feel it is important to include the simple regression as well as the multiple regression results. I would be very grateful if you could re-instate the deleted Table 3.

I sincerely apologise for any confusion caused by the reviewer comment - I think it was clarification that was sought, rather than removal of the table.

Many thanks for your understanding and I look forward to seeing your resubmission.

With best wishes,

Gracia Fellmeth

(Guest editor)

---

## [Author Response · Author response to Decision Letter 2]

8 Jan 2021

Response to comments from the Editor: 

I ask if you can make one final change: in response to one of the reviewer comments, you removed what was previously Table 3 (simple linear regression). In the previous version, it was not clear that Table 3 was indeed a simple regression as it appeared that other factors were being controlled for. However, given your response and clarification, I feel it is important to include the simple regression as well as the multiple regression results. I would be very grateful if you could re-instate the deleted Table 3.

Thank you for your suggestion. We have re-instated Table 3 (simple linear regression) in the revised submission, and we have slightly modified the text to make it clearer.

Page: 14 Line: 215-217

"Table 3 shows the results of simple linear regression analysis for the association of each independent variable with psychological well-being of adolescents, without controlling for other independent variables."

Page: 15 Line: 225-227

"Table 4 presents the results of multiple linear regression analysis for the association of each independent variable with psychological well-being of adolescents, while controlling for other independent variables."

---

## [Editor Report · Decision Letter 3]

11 Jan 2021

Parental migration and psychological well-being of left-behind adolescents in Western Nepal

PONE-D-20-14440R3

Dear Dr. Ong,

We’re pleased to inform you that your manuscript has been judged scientifically suitable for publication and will be formally accepted for publication once it meets all outstanding technical requirements.

Kind regards,

Gracia Fellmeth

Guest Editor

PLOS ONE
---

## [Editor Report · Acceptance letter]

20 Jan 2021

PONE-D-20-14440R3 

Parental migration and psychological well-being of left-behind adolescents in Western Nepal 

Dear Dr. Ong:

I'm pleased to inform you that your manuscript has been deemed suitable for publication in PLOS ONE. Congratulations! Your manuscript is now with our production department. 

Kind regards, 

on behalf of

Dr. Gracia Fellmeth 

Guest Editor

PLOS ONE